# This bookmark gauges the depths of the human: how poetry can help to personalise climate change

*Sam Illingworth*[1]
[1]*School of Science and the Environment, Manchester Metropolitan University, Manchester, UK*

## Abstract

By conducting a qualitative content analysis of 72 poems written about climate change by poets from across the world, this study demonstrates how these poets have interpreted the, at times, esoteric principles of climate change. The results of this study indicate that these interpretations highlight the need to re-position humans in the epicentre of the debate so that a meaningful dialogue around the subject might be established, especially amongst non-specialists.

**Correspondence:** s.illingworth@mmu.ac.uk

## 1. Introduction

For each of the last three decades, temperatures at the Earth's surface have been rising, reaching levels higher than any recorded since the middle of the nineteenth century, when multiple independently produced measurements first began (Stocker et al., 2013). This recent warming has been caused by an anthropogenic increase in the atmospheric concentrations of carbon dioxide, methane, and other greenhouse gases, which have increased to levels unprecedented in the last 800,000 years (Seinfeld and Pandis, 2016). Carbon dioxide concentrations alone have increased by 40% since pre-industrial times, primarily from fossil fuel emissions and secondarily from emissions caused by changes in land use (Leung et al., 2014). Understanding and quantifying greenhouse gas emissions is central to international efforts to slow their growth rate in the atmosphere, in order to mitigate the humanitarian and economic impacts of global warming.

The effects of increased greenhouse gas emissions are not just limited to an increase in global temperatures; they are also profoundly influencing our climate, resulting in an increase in the number of heatwaves, extreme weather events and flood risk (Van Aalst, 2006). However, the implications of climate change on our environment and society is not solely dependent on how the Earth system responds to changes in greenhouse gases; instead it depends on the extent to which humankind responds through changes in their lifestyle, attitude, and policy (Moss et al., 2010). Therefore, alongside the work of scientific research that aims to quantify these emissions (see e.g. Palmer et al., 2018), it is necessary for non-scientists to support and develop appropriate mitigation strategies against global warming. In order for this to be done effectively, they need to be both aware that it is taking place, and to be certain that it is anthropogenic (Hassol, 2008). They also need to realise that no matter where they are in the world they are at risk from the effects of climate change (Dominelli, 2011).

Howe et al. (2015) conducted a study amongst US citizens to determine the extent to which they believed global warming was happening, and how they believed it affected them. They found that of the 12,061 people surveyed between 2008 and 2013, 70% believed global warming to be happening, while only 53% believed it to be anthropogenic. Similarly, only 49% of them considered scientists to agree on the anthropogenic nature of global warming; in reality that consensus is at least 97% (Cook et al., 2016). Amongst these same participants, a slim majority (51%) believed that global warming was already harming people in the US, yet only 40% thought that global warming would harm them personally, with 33% of respondents stating that they discussed global warming at least occasionally with friends and family. These results would therefore suggest that while many US citizens still need convincing about the anthropogenic nature of global warming, a more pressing concern is perhaps the need to convince them of the risk that it poses at the individual and local level.

It is perhaps unfair to single out US citizens for such analysis. Between 2008 and 2009, Gallup (the global performance-management consulting company) conducted a major

worldwide poll across 127 countries about personal attitudes towards climate change (Gallup and Newport, 2010). While this is an older data set, the results are in-line with the work of Howe et al. (2015): 63% of people surveyed claimed to know something about climate change, with only 55% agreeing that it was anthropogenic and 47% acknowledging that it posed a serious personal threat. While many climate change communication efforts focus on convincing citizens of the anthropogenic nature of climate change (see e.g. Nerlich et al., 2010), more work is clearly needed to help address the perceived disconnect between global effects and personal threat. What is needed is something that can transcend cultural barriers, and which can contextualise and personalise a global problem. What is needed is poetry.

In his treatise *A Defence of Poetry*, the English Romantic poet P.B. Shelley (1890, pp. 46) wrote that:

> Poets are the hierophants of an unapprehended inspiration; the mirrors of the gigantic shadows which futurity casts upon the present; the words which express what they understand not.

A hierophant is considered to be a person who interprets sacred mysteries or esoteric principles. Is there a mystery more sacred than how best to safeguard our planet? Is there a principle more esoteric than the effective mitigation of climate change? In Ancient Greece, hierophants were needed to interpret the will and needs of the gods for the rest of society; at the behest of Shelley might we now turn to poets to interpret the will and needs of our planet? Talking about climate change is difficult. Even experts find it challenging to establish a common language that communicates their research, statistics, and emotions effectively (see e.g. Hulme, 2009). Poetry offers a way to establish this common language, presenting an opportunity for people to express themselves in a different way, to find a fitting language that enables them to talk about climate change in a manner that is personal to them, and which can potentially help them to find the words that are needed to communicate with others more effectively (see e.g. Illingworth and Jack, 2018 and references therein).

By conducting a detailed qualitative content analysis for a selection of climate change poetry, this study aims to understand how poets have interpreted the principles of climate change, and how these interpretations might be used to engender the dialogue that is needed to meaningfully address the issue. In Section 2, I discuss the methodology that I adopted in this study, and in doing so outline a new approach with regards to how poetry might be used as data to reveal insight into a particular topic (in this instance attitudes towards climate change). Section 3 contains a discussion of how the emergent categories and themes relate to the research questions, and Section 4 contains the conclusions, along with future directions for research.

## 2. Methodology

The methodology that I adopted in this study involved treating poetry as data, allowing for a contextual meaning of the text to be analysed in relation to climate change. While several other methods exist for the analysis of textual data (e.g. ethnography, phenomenology, grounded theory, etc.), I have chosen qualitative content analysis because of its ability to highlight both the context and the content of the chosen text, which for a subjective medium such as poetry is essential. In outlining the methodology that was used in this study I also intend to provide a blueprint for the qualitative content analysis of poetry with respect to other topics of interest. Previous studies have treated poetry as data to explore certain topics but have tended to focus on methods of inquiry (see e.g. Furman, 2004;Hunter, 2002;Shapiro, 2004), autoethnography (see e.g. Furman, 2006;Maurino, 2016), or quantitative coding (see e.g. McDermott Jr and Porter, 1989;Hoover et al., 2014). Similarly, while other research has been conducted in relation to climate change and poetry, this has tended to focus on either literary criticism (see e.g. Trexler and Johns-Putra, 2011;Griffiths, 2017) or action research (see e.g. Miller and Brockie, 2015), the former of which typically involves re-reading much older bodies of texts, while the latter introduces recall and interviewer / facilitator bias. By performing a qualitative content analysis on poetry that has been written recently, but not for the sole purpose of research, this study aims to better understand the way in which poets interpret climate change, and how this might be used to better personalise the subject.

Any approach which utilises a qualitative content analysis should be guided by these seven steps: formulate research questions; select sample to be analysed; define the categories to be applied; outline the coding process; implement the coding process; determine trustworthiness; and analyse the results of the coding process (Hsieh and Shannon, 2005). In defining my methodology, I will outline the first six of these steps here, with the seventh (the analysis) being presented in Section 3.

### 2.1 Formulation of Research Questions

As discussed above, the combination of poetry as data and qualitative content analysis as method were chosen so as to better understand the ways in which poets independently interpret the principles of climate change, and in doing so how this might be used to widen the debate around climate change by making it something that people identify more personally with. For the purposes of this study, this was formalised into the following two research questions:

> RQ1: how have poets interpreted the, at times, esoteric principles of climate change?
> RQ2: how might these interpretations be used to better personalise the debate around climate change so that it is discussed more widely?

### 2.2 Selection of Samples to be Analysed

In selecting the poetry for this study, I wanted to engage with a body of work that captured a wide range of interpretations, and from a large number of poets. Selecting poetry from only one or several poets would have limited the potential interpretations, while picking poetry which I identified as being about climate change could potentially have introduced an interpretative bias before any content analysis had taken place. As such I needed a collection of poetry that was definitely about climate change, and which was written by more than a handful of poets. At this stage I also decided to rule out any venture that I had personally been involved with (either through the editing, soliciting, or submission of poetry) so as to avoid interviewer / facilitator bias.

*Magma* is an international magazine of poetry that is published three times a year in Spring, Autumn and Winter, both on paper and as a digital edition. The editorship circulates among the group which runs the magazine, with an occasional guest editor, and the ethos of the publication is a commitment to publish the best in contemporary poetry, from little known poets to more established ones. Each issue has a designated theme, with submissions for each issue released several months before. Issue 72 of *Magma* was entitled 'The Climate Change Issue', with the following call for submissions advertised via their website (Magma, 2018):

> We're looking for poems that engage with the theme of climate change in any way, that reflect it, have it as an emotional underlay, or react against it… Send us poems of grief, anger, despair, dystopian angst, scepticism, devil's advocacy, activism, optimism, humour, joy… Elegies, satire or whatever.

The openness of the call made it clear to the poets that they were free to interpret the topic of climate change, which made it an ideal data source for this study. In addition to an editorial, book reviews, and extended features 'The Climate Change Issue', which was published in Autumn 2018 and edited by Matt Howard, Fiona Moore, and Eileen Pun, featured 72 original pieces of poetry from 57 authors (Howard et al., 2018). The background of the poets was considered, but only after the coding had been done so as to avoid any bias. After reading the biographical information of these poets and conducting a background search, only two of them could be considered to be active scientists, one of whom is a futurist working for a sustainability non-profit organisation, and the other of whom is an environmentalist, who at the time of writing was working on a master degree in Ecology and Environmental Studies. Given that the RQs are focussed on how poets have interpreted climate change for a non-specialist audience, and that both of these writers self-identify as poets, their poetry was not excluded from study, especially since the ideas and themes explored in their poetry did not result in the emerging of any new codes or categories (see Section 2.4). In addition to the inclusion of these two scientist poets, several of the poems in the issue (8 in total) came about from invited discussions between scientists and conservationists from the Cambridge Conservation Initiative. However, the poets themselves could still be considered to be non-specialists who were interpreting climate change following conversations with climate change experts, and so their poetry was included in the analysis.

While it is not necessarily the case that poetry anthologies will always exist for a particular topic, it is also true that many poems do in fact make the topics of their intent sufficiently clear so as to avoid interpretive bias. However, in order to answer RQ1 for this study it was necessary to pick contemporary poetry written from a wide selection of poets, for which 'The Climate Change Issue' presented the ideal source. The following quotation, taken from the editorial, also outlines how the overarching tenet of this issue is fully congruent with the rationale behind this study, i.e. that climate change should not be just the sole preserve or concern of the scientist (Howard et al., 2018, p. 5):

> It seems redundant to say climate change isn't just a scientific concern when its scope is no less than total – perhaps we are waiting for human consciousness and behaviours to catch up.

## 2.3 Definition of Categories to be Applied

A conventional approach to qualitative content analysis was adopted in this study, with pre-conceived categories being avoided, and instead being determined by the implementation of the coding process (see Section 2.4). While in some instances a directed content analysis might be more appropriate, this is usually used in those instances where an existing theory would benefit from further description (Hsieh and Shannon, 2005). As the research questions to be addressed in this study are unique, a directed approach is inappropriate. Similarly, a summative content analysis would fail to fully account for the context of the poetry alongside its content.

## 2.4 Outline and Implementation of Coding Process

The outline and implementation of the coding process have been combined here, as they are closely interrelated, and discussing them together serves to better highlight how such an approach was adopted in this study.

A traditional approach to coding data during qualitative content analysis would be to begin by identifying meaning units in the text, condensing these down to smaller units and then labelling these units with codes. These codes would be chosen so as to describe what each meaning unit was about, after which different codes would be grouped into thematic categories according to content and context, before looking for any emerging theme(s) that expressed an underlying meaning of the text and which could be directly related back to the research question(s) (Erlingsson and Brysiewicz, 2017). Whilst this overall schema can be observed in the process outlined below, the approach that I adopted differed slightly in its treatment of condensed meaning units, which should be avoided when treating poetry as data for qualitative content analysis. This is because in addition to overly short meaning units leading to fragmentation (Greneheim, 2004), poems, unlike transcripts or survey responses, have been crafted by the author so that every word and sentence has 'meaning'. As such each

line (and perhaps each word) of the poem could already be considered to be a meaning unit and should not be condensed further.

In conducting my analysis, I began by reading all of the poems in 'The Climate Change Issue' to familiarise myself with their content and context. I then went through each of the poems in the order in which they appeared in print, and assigned codes to sections of the poems that addressed RQ1 (i.e. how had these poets interpreted climate change). Assigning an overall meaning or tone to the poem as a whole was avoided, as this would introduce a degree of subjectivity that is inappropriate unless a phenomenological approach is being adapted, in which the lived experiences of the researcher(s) is being considered as an essential part of the analysis (see e.g.Illingworth and Jack, 2018). As such an approach is not compatible with the research questions of this study, I instead assigned codes to lines of text which made reference to a specific label. These labels emerged from the poems, and were chosen to be as objective as possible, as can be seen from Table 1.

As well as avoiding tone and sticking to specific references in the text, coding occurrences were always chosen to be literal rather than metaphorical or symbolic, so that further subjectivity could be avoided. For example, "and gulls strewn like heaps of soiled rags among oil-glistened // bodies of harbor seals after the blowout on Platform A" was coded as 'Fauna', whereas "I meet Al Gore // in the lovely woods // of sleep // he's braver // than a tiger" was not, as in this instance the tiger was being used to symbolise bravery (here, and throughout this manuscript, // is used to indicate a line break in the poem, i.e. the termination of one line of the poem and the beginning of a new one.). These lines were however coded as 'Humans' because they made explicit reference to a human being other than the author of the poem, i.e. Al Gore.

As each new code was realised I went back through the poems that had previously been coded to see if these also contained any lines that could be labelled with this newly emergent code. I then read all of the poems in full again and made sure that each of them had been coded accurately and that a saturation of emergent codes had been reached. This resulted in a total of 21 codes. I then read each of the poems again and made sure that no coding had been missed. Following this I went through each of the individually coded segments and checked to make sure that they really did belong in this category, checking that (for example) Al Gore being described as a brave tiger was coded as 'Human' rather than 'Fauna'. At this stage I realised that one of the codes that I had created was at odds with my methodology, and so it was removed. 'Personification' has been defined as 'any poems that were written as if from the point of view of nature / the Earth system', and although there were four such instances of this code, I considered this to be too subjective for the analysis, and so it was removed. This resulted in the 20 codes that are outlined alongside their definitions in Table 1.

After this coding had taken place, I read through all of the coded references and then grouped these into categories, which consisted of codes that appeared to deal with the same issue. Table 2 outlines the categories and corresponding codes, along with the number of times they occurred. These categories, and their relation to the research questions are discussed further

in Section 3. After these codes had been grouped as such I went back through each of the individual occurrences (e.g. the 152 segments of poetry that were categorised as 'Habitat') to make sure that they did indeed belong in this category. As can be seen from Table 2, this resulted in 5 individual categories: 'Habitat', 'Reactions', 'Language', 'The Present', and 'Our Future'.

Following this categorisation of the codes, they were further examined for any themes that expressed underlying meaning in relation to the research questions (Erlingsson and Brysiewicz, 2017), the results of which are presented in Section 3.6.

### 2.5 Trustworthiness of Coding

In order to improve the trustworthiness of this content analysis, I followed the checklist outlined by Elo et al. (2014), which involved checking for trustworthiness at the preparation, organisation, and reporting phases of the analysis. In the preparation phase, the data collection, sampling strategy, and unit of analysis (unit of meaning) selection were carefully considered and have been justified above. During the organisation phase, the categorisation, interpretation, and representativeness of the analysis was assured by repeatedly checking for consistency, e.g. by checking each of the individual occurrences of text against the categories. The reporting phase is covered in Section 3 of this study, but here trustworthiness was assured by providing enough detail to ensure that the reader can evaluate the transferability of the results.

In order to establish the trustworthiness of the analysis of poetical data, Shapiro (2004) also recommends establishing an audit trail, ensuring that there has been a theoretical saturation of the data, and where possible involving more than one researcher. While the audit trail and saturation of data have been discussed (with Table 1 and Table 2 demonstrating how the emergent codes and categories in this study were defined and organised), in this instance only one researcher was used to analyse the data, and as such this may introduce biases to the interpretation of the data. However, this is also true for any content analysis that involves only one researcher (Elo et al., 2014). As the goal of this analysis is not to guarantee the systematic development and use of a code book, the interpretive process is not overtly affected by the use of a solo researcher. Furthermore, the transparency of the coding and subsequent analysis further improves the trustworthiness of the approach.

## 3. Results and Discussion

As can be seen from Table 2, five major categories emerged from the methodology that was adopted in analysing these poems. I now discuss each of these emergent categories, how they relate to RQ1 ("how have poets interpreted the, at times, esoteric principles of climate change?"), and how they compare to other research that has been conducted in terms of the communication of climate change. Following a discussion of these categories I present the overall theme that emerged from conducting this analysis, and how this relates to both RQ1

and RQ2 ("how might these interpretations be used to better personalise the debate around climate change so that it is discussed more widely?").

### 3.1 Habitat

The most prominent category to emerge with regards to the ways in which poets interpreted the principles of climate change was 'habitat'. This category emerged from a variety of different sources, with many of the poems focussing on a celebration of habitat (either the flora or the fauna or both) as is evident from the snippets of the following two poems: 'A Trip to Mount General in Late Winter' by Huang Fan and translated from Chinese into English by Lei Yanni (Howard et al., 2018, p. 13):

> In the bamboo grove where you can almost
> forget who you are – if you are steadfast as the plum blossoms
> that hold on to early spring

And 'Beijing Parakeets' by David Tait (Howard et al., 2018, p. 11)

> but I wait beneath the bare pomegranate tree
> and watch the two old parakeets, lovebirds,
> huddled up together, one cleaning the feathers
> on the other's head, the other softly singing.

Both of these poems celebrate habitat, but they also ground this celebration in how habitats (and nature) are experienced and appreciated by humans, as is also evident from this extract from 'Notes from a transect' by Polly Atkin (Howard et al., 2018, p. 47)

> One school wins a visit from the scientist. When she asks
> *does anyone have wildlife stories to share?*
> the whole school put up their hands.

In contrast to this celebration of current habitats, and how they are appreciated, several of the poems also considered the loss of habitat. The following two extracts from 'An eco-worrier tweets' by Neetha Kunaratnam (Howard et al., 2018, p. 41) and 'ISOTHERM' by Jos Smith (Howard et al., 2018, p. 54), demonstrate how this loss was explored by the poets for both flora and fauna, respectively:

> while we pine for the pines,
> and they plane the mighty planes

And:

> What does a loss of birds look like?

> What is the collective noun
> for such losses? Would you hear
> the silence of lapwings, of thrushes?

As with the celebration of habitat, what is particularly interesting with regards to how the poets chose to represent this loss, was that it was almost always contextualised with respect to humans, i.e. “*we* pine for the pines” and “Would *you* hear the silence of lapwings” (emphasis in italics is my own). While the following extract from ‘Notes from a transect’ by Polly Atkin (Howard et al., 2018, p. 48) makes clear that this habitat loss should not be ranked, it is clear that any quantification / rationalisation of loss is seen by the poets to be reliant on human consideration:

> Is it cheaper to weep for a sea otter – clutching
> paws in the water – than a lake?

Exploring this idea of loss further, it is the relationship between humans and habitat, and in particular how conflict has arisen to become the dominant connection between the two, that many of these poems aspire to, as is evident from this extract from ‘The loss of birds’ by Nan Craig (Howard et al., 2018, p. 64):

> They were everywhere, I insist. *Everywhere*.
> You smile politely and begin to drift away.
> WAIT! I shout. They also *sang*!

This need for human contextualisation might be seen to be an unconscious (or conscious) reflection by the poets on the role that humans are playing on impacting the climate, and the fact that we are the only species that are able / willing / conscious of making such an impact. This concept is further evident in Matthew Griffiths’ ‘Pantones for the Anthropocene’, the very title of which makes reference to the current geological epoch, viewed as the period during which human activity has become the dominant influence on climate and the environment (Howard et al., 2018, p. 35):

> This bookmark gauges the depths of the human,
> Laid to the layers to show where a new one
> Rises like icing, a fresh fall of snow on
> A stiffening stratum, and so – with the golden
> Spike on the graphlines not otherwise seen –

Habitat loss, and in particular extinction risk, has long been presented by scientists as one of the most visible effects of climate change, with e.g. Thomas et al. (2004) stating that a large fraction of species could be driven to extinction by expected climate trends over the next 50 years. As such, it is perhaps not surprising that many of the poets chose to explore the role of habitat and climate change, and in doing so further examine the evolving relationship between humans and nature. What these poems make evident however, is that despite our

behaviours (and the original code that was adopted in Table 1) it is impossible to view ‘humans’ and ‘nature’ as two mutually exclusive entities, as although anthropogenic climate change may be having a hugely negative effect on nature the two systems are clearly interrelated, or as noted by Corlett (2015, p. 4):

> If humans are now the dominant ecological force on the planet, then it is impossible to separate ‘humans’ and ‘nature’ in the way that conservation has traditionally tried to do.

### 3.2 Reactions

This category represents those poems that explore the reactions that humans have towards climate change, the largest proportion of which represent an acknowledgment that climate change is happening and also that humans are largely to blame for its cause and effects, either because of very specific incidents, as evidenced in this extract from ‘Río Nuevo’ by Leo Boix (Howard et al., 2018, p. 75):

> New owners didn’t rotate their crops.
> A Martian landscape rapidly arose.

Or because of more general attitudes and behaviours, as expressed by Patrick Sylvain in ‘Ego’ (Howard et al., 2018, p. 26):

> In the boundless universe,
> I am less than a speck.
> But my ego,
> The size of a planet,
> Defames the world.

The outcomes of these attitudes are also examined by the poets, with Matthew Griffiths, in his poem ‘Pantones for the Anthropocene’, exploring the notion that burying our heads in the sand has simply served to further distance ourselves from both the problem and also nature more generally, (Howard et al., 2018, p.35):

> Lifting our arses up in the confusion
> Of air and ourselves we have made of the future
> And off the hot core of that gobstopper, nature.

Alongside this general acknowledgment that climate change is anthropogenic, there is also some doubt. However, this reaction does not manifest itself in terms of climate change denial, but rather in terms of the degree to which we can truly quantify its extent, as demonstrated by Penelope Shuttle in ‘An Inconvenient Truth’ (Howard et al., 2018, p. 65) :

> no one knows where the past goes

no one knows anything about
anything on this dirty little planet
of ours

This doubt and uncertainty is accompanied by a realisation that climate change is not a simple problem, either in conception or communication, as Polly Atkin observes in 'Notes from a transect' (Howard et al., 2018, p. 46):

in the data        the scientist says        it's hard
to ask questions        to prise apart    correlation
habitat or climate        disturbed or not
disturbed        perception or preconception
it depends what scale you concern yourself with

An interesting issue that arises in these poems is that despite an acknowledgment and ownership of the problem, very few solutions for mitigating against or even adapting to climate change are presented. In 'A way of managing diversity' Kathryn Maris tells us that "We must band together against this encroaching threat" (Howard et al., 2018, p. 58), while in 'Do not turn this page !!!' Roger Bloor states "then what is the answer? // 0 level carbon emission target" (Howard et al., 2018, p. 98). However, despite a lack of actual solutions several of the poets still express hopes for the future, with Joanna Guthrie observing in 'Here, afterwards' that (Howard et al., 2018, p. 12):

at which you will look down
from time to time
amazed at the journey
their new strength
the way that they've
adapted best of all
to this time

In considering the reactions that humans take towards climate change, these poems have interpreted climate change as something that does exist, and that we (as humans) are largely to blame for, but there is a distinct lack of any real, or even perceived, solutions to the problem. There is hope, but less certainty in what this will actually look like / how it will physically manifest itself. There is also an acceptance that things are not simple, and that in interpreting these results and trying to make sense of them, scientists have a difficult job that is made more so by trying to represent error bars and standard deviations as something that still possesses an urgency. Such an attitude is reflective of recent research that has revealed that the language used by the global climate change watchdog, the Intergovernmental Panel on Climate Change (IPCC), is overly conservative (Herrando-Pérez et al., 2019).

Previous studies (see e.g. Budescu et al., 2009) have shown that there is a large disconnect in the ways that scientists and non-scientists understand uncertainty, and that as such the communication of uncertainty has the potential to undermine effective action unless climate change messages are framed appropriately (Morton et al., 2011). However, these poems would seem to suggest that the poets take into consideration the nuances of quantifying climate change. These poems also clearly demonstrate that there is an acknowledgment of the anthropogenic nature of climate change, but that a likely barrier to engagement is a perceived lack of potential solutions, as has also been discussed by e.g. Lorenzoni et al. (2007).

### 3.3 Language

Another category to emerge from this content analysis was the importance of language. Many of the poems adopted language that could be considered to be spiritual or quasi-religious; for example, Ben Smith in the poem 'Data Sets' observes that (Howard et al., 2018, p. 18):

> This is the real work of divination:
> not grand prophecies
> but data gathering.

While 'Data Sets' uses quasi-religious language as a comparison for the underlying science of understanding climate change, several other poems encompass this form of language as a direct invocation for protection and/or help from a higher power, as is evident in these lines from Sarah Gridley's 'Diabolic Clouds Over Everything' (Howard et al., 2018, p. 97): "For the love of God, // or otherwise", and also these from Leo Boix's 'Villanelle (Un Paisaje)' (Howard et al., 2018, p. 9): "An altar to pray for a better world".

In contrast to this use of spiritual language, other poems use a form of language that could be classified as scientific, i.e. they make reference to a specific fact or piece of technical jargon, such as the line 'Light breeze is the first sign of barometric change' in Rachel Mead's poem 'A Beaufort Scale for Depression' (Howard et al., 2018, p. 28) or "Say hello to the Man Age, so long to the Holocene" in Matthew Griffiths' 'Pantones for the Anthropocene' (Howard et al., 2018, p. 35), where the poet explains the title of the poem by making reference to another geographical period, and drawing attention to the notion that the Anthropocene is a functionally different epoch from that of the Holocene (see e.g. Waters et al., 2016). By using scientific language in this way, the poets are introducing their readers to new research and findings albeit in a markedly different style to that used in scientific research or even popular science articles.

One of the most stylistically interesting poems in the collection is Cat Campbell's 'CH4 is a much more potent greenhouse gas than CO2', which takes the work done by Worrall et al. (2010) on 'Peatlands and climate change', and interspaces the scientific findings of this report with lines of poetic text (represented in italics), the effect of which is to introduce the reader to scientific fact (both that of the title and the notion that peatlands can be a source as well as a sink of carbon) whilst simultaneously humanising it (Howard et al., 2018, p. 15):

It is possible for a peatland,
*site of battles and back-breaking crofting,*
to be a net sink for carbon,
*blood, sweat, grief and hate,*
but at the same time
*to be a source of enough tranquillity*
to have a net positive
*effect on human nature and a*
radiative forcing (i.e., warming)

As well as turning to the languages of science and religion in an attempt to convey their message, several of the poems also made use of languages other than English. The poems in this collection included only one complete translation, '暮冬时节将军山行' by Huang Fan that was translated from Chinese into English as 'A Trip to Mount General in Late Winter' by Lei Yanni. The other poems that used a language other than English interspersed the text with words from that language, such as the use of Spanish by Leo Boix in Villanelle (Un Paisaje)' or 'Stotterin inta Anthropocene' by Christine De Luca, which was written entirely in the Shetlandic dialect, with the reader not presented with a translation, but rather a glossary of terms (for example, that the word 'glunsh' means to 'swallow greedily'). What was particularly interesting about these poems was that the author had clearly chosen to write sections of the poem in a language other than English as it enabled them to more fully express what it was that they meant to say about climate change.

In considering the emergent category of language across these poems, it is evident that using only a singular official language (i.e. English) or technical language (i.e. science) is not sufficient to interpret and communicate the causes and consequences of climate change, and that by doing so we are at risk of ostracising those communities that are not fluent in these chosen languages. English-speaking status has been shown to be a limiting factor in participating in the IPCC (Ho-Lem et al., 2011), whilst many studies often omit non-English research when conducting large-scale research into barriers to climate change adaptation (see e.g. Biesbroek et al., 2013). These poems suggest that by restricting the *lingua franca* of climate change to scientific English, it is perhaps not surprising that it is discussed less widely than is needed for meaningful action to take place.

### 3.4 The Present

This category considers those poems that make reference to the current state of the climate change system, outside of those already emergent in the category of habitat discussed in Section 3.1. Poems that were categorised as such included those that discussed the weather as an interrelated aspect of the climate system, either through a specific example, as demonstrated in this extract from 'Change' by D A Prince (Howard et al., 2018, p. 29):

But these fields are,
again, under water, brought
to the brink of drowning

Or else through the notion that something is 'not quite right', and that one of the ways that this can be observed is through changes in the weather, as is apparent in 'This Weather' by Siún Carden (Howard et al., 2018, p. 29):

she finds it swirling there, and she can't say
she's been herself, this weather.

In addition to the current state of the weather, this category also considered those poems that made reference to the current state of pollution. The majority of poems that made reference to this topic were concerned with plastics in the oceans, such as this extract from 'There is No Alternative' by Momtaza Mehri (Howard et al., 2018, p. 56):

the future belongs to the yolky bopping heads of plastic ducks
green bottle caps cigarette butts everything touched by the lips
then cast unuttered into oceans into the pooled memory cells of the universe

There was only one mention of air pollution in any of the poems, occurring in 'Beijing Parakeets' by David Tait: "I've already got a pollution headache … the smog of Beijing simmering around us." (Howard et al., 2018, p. 11) The relative popularity of plastic pollution in these poems is likely symptomatic of the increase in public attention that this issue has received following the BBC TV series *Blue Planet II* and the subsequent media outcry (see e.g. Kenward, 2018). In future years, such a collection of poetry might would likely contain more poems on other environmental topics that had risen amongst the public consciousness.

Across all of the poems, only two of them made reference to an actual historical event and in both instances, these referred to storms. In 'Howling Wind', Patrick Sylvain observes how "Hurricane Matthew broke spines already fractured" (Howard et al., 2018, p. 26), while in 'Tip #5 What not to say whilst online dating', Helen Moore recalls a recent storm in Bristol, remarking that (Howard et al., 2018, p. 60):

Beaufort 9 bludgeoning      Bristol, pounding the city

like WWII was recurring. On the Harbourside,

gales chucking slops at houseboats, yachts,
clinking masts like Chinese businessmen gan bei-ing a deal

It should be noted that while one of these poems recalls a well-known global event (Hurricane Matthew was the storm that caused catastrophic damage and a humanitarian crisis

in Haiti in the Autumn of 2016) and localises it to the frame of reference of the reader, the other makes reference to a localised storm and contextualises it with reference to a global event (WWII), thereby highlighting the ability of the poet to interpret and frame the principles and effects of climate change in order to communicate to the reader.

The poems in this category also consider the general effects of climate change in terms of things being either broken or killed, not in terms of specific fauna or flora (see Section 3.1) but rather a general sense of death and destruction, as evidenced by the following line from 'Beaufort Scale for Depression' by Rachel Mead (Howard et al., 2018, p. 28): "Widespread structural damage. Zero visibility. This is the point of collapse, the black hole."

This category highlights the 'messy', interrelated nature of climate change, and demonstrates that poets are not afraid to discuss several different systems (climate change, weather, pollution, etc.) in order to communicate to their audience. While scientists are often at pains to point out the differences between weather and climate, and the confusion that such a misunderstanding can entail (see e.g. Weber and Stern, 2011), it is also true that beliefs in climate change are affected by local weather conditions (Li et al., 2011). By presenting changes in both the weather and climate alongside one another, the poets are aiming to reach out to their audience and ground them in a language that they understand rather than to confuse them or cut off from a particular line of enquiry. By not allowing such interrelated discussions to take place (confusing as the may sometimes be), there is also the argument that a non-scientific audience is being denied access to solutions from an interrelated filed. One such example is the success of the Montreal Protocol in tackling the Ozone Layer (Oberthür, 2001), as while it has been shown that a non-scientific audience often confuses stratospheric ozone depletion with the greenhouse effect (Bostrom et al., 1994), presenting the Montreal Protocol as an exemplar of how government policy can engender positive environmental change on a global scale, can help to present some of the potential solutions to the climate change issue that these poems have highlighted as being less than readily available (see Section 3.2), thereby overcoming one of the potential barriers to dialogue.

### 3.5 Our Future

In contrast to the previous category, this final category is one that emerged as a result of poems that discuss possible futures that might arise as a result of the current climate system. There is a large range of temporal scale in these poems, with some imaging the fallout of a climate catastrophe in a not-too-distant future, such as that presented in this extract from 'There Is No Alternative' by Momtaza Mehri (Howard et al., 2018, p. 56)

> The Alliance of Small Island States were the earliest to disappear
> the shepherds were the last the gospel preachers of accumulation had nowhere to go
> they were too busy competing with the skies to notice them folding in

Whilst others are grounded in a future quite markedly different from our current state, such as 'Theft-saving' by Amaan Hyder, who imagines a future where (Howard et al., 2018, p. 63):

> You fly a distance of twenty planets
> to a zoo to see your first animals,
>
> pure as the night their ancestors were taken,
> beamed up out of extinction.

And others much further still, with 'I was human once' by Ama Bolton considering the Earth system many years from now when there are no humans left at all (Howard et al., 2018, p. 8), and where:

> through centuries of firestorm
> when things cool down          I'll know it's time
>           to spin the whole unholy yarn
>            all       over      again

Whilst these poems create the framework for a future Earth based on a variety of different scenarios, other poems also reflect on the 'consideration of the future' itself, and how useful (or not) this might be in combatting climate change. This extract from Sarah Gridley's 'Diabolical Clouds Over Everything' being a particularly powerful rallying call against the inaction that can sometimes arise from over-pontification (Howard et al., 2018, p. 97):

> No one will draw in the future. Soon
> we will stop having to ask,
>
> What must the future hold?

Aside from discussions of imagined futures for the Earth system and humans in general, the poems in this category also make specific reference to children and their relationship with both ourselves and nature. Some of these poems focus on what we choose and have chosen to leave behind as an inheritance, such as in 'Estate' by Steve Kendall (Howard et al., 2018, p. 96):

> To our children
> we bequeath the promises we made, their rightful solitude

Other poems consider the responsibilities that we have for our children's current and future wellbeing, as evident by the line "I would like my children to feel safe" in Kathryn Maris' 'A way of managing diversity' (Howard et al., 2018, p. 58). By asking the reader to consider the future implications of climate change on future generations these poems support the narrative that many members of the public consider providing a better life for future generations to be the most important motivator in taking action against climate change (see e.g. Leiserowitz et al., 2009). As noted by Pahl et al. (2014), in order for people to acknowledge the future implications of their current lifestyles and community choices, it is first necessary to improve

how we engage them in envisioning the future, and as is demonstrated here poetry provides one potential way for providing this engagement.

### 3.6 An Emerging Theme

In considering these categories in the context of RQ1 ("how have poets interpreted the, at times, esoteric principles of climate change?"), a clear theme emerges: the central role that is occupied by humankind. This role concerns how we as humans have accepted our past, how we are moulding our future, the extent to which we are defending and destroying our shared habitat with nature, and how we determine both the language of communication and appropriate reactions.

This positioning of humans in the epicentre of the climate change debate might at first be seen to be somewhat egotistical or even narcissistic. Just as the famous philosophical thought experiment asks 'if a tree falls in a forest and no one is around to hear it, does it make a sound?' to some extent these poems ask us to consider 'if the climate is changed but no one is around to measure it, does it actually change?' There is an arrogance here, but in addressing RQ2 ("What does this tell us about how scientists can talk about climate change to non-specialist audiences?") it is a necessary one, i.e. that in order to establish the dialogues that are needed to enact change it is vital to remind audiences of the central role that humans *do* occupy in terms of both cause and effect. Without this re-positioning, there is a danger that climate change will be assumed to be beyond the control and responsibility of humankind; yet, as noted by Urry (2015, p. 46) it is vital to remember that climate change "is not a purely 'scientific' problem and that human actions are central to this apparent warming of the planet." Similarly, without such re-positioning the phrase 'climate change' itself risks being interpreted as a phenomenon that is passively happening, rather than something that we, as humans, are both causing, and are thus ultimately responsible for mitigating.

Whilst studies such as those conducted by O'Neill and Nicholson-Cole (2009) have shown that fear is generally an ineffective tool for motivating genuine personal engagement, failing to remind people of the role that humans have played in causing climate change, and the role that they must now assume in mitigating against it, is arguably equally ineffective in establishing the dialogue that is first needed before meaningful action can take place. In the foreword to the poem 'Sample Basket Red List 2318', Jen Hadfield writes that (Howard et al., 2018, p. 68):

> To approach the global crisis we need to attend to the local crisis. Isn't approaching the global crisis by addressing local specificity one of the things poetry is best at?

By acting as modern-day hierophants, this study argues that poets can highlight to scientists and communication experts the challenges to engendering individual and collective action on the topic of climate change. These findings manifest themselves in a need to re-position humans at the centre of the climate change debate, and in so doing to consider the use of a language that is localised and personal, to help broaden the conversation to every human.

## 4. Conclusions

By acknowledging that there is a lack of dialogue around climate change amongst a non-specialist audience, this study set out to ask “how have poets interpreted the, at times, esoteric principles of climate change?” (RQ1) and in doing so to determine “how might these interpretations be used to better personalise the debate around climate change so that it is discussed more widely?” (RQ2). By conducting a detailed qualitative content analysis on a selection of climate change poetry, a number of categories emerged with regards to the poets’ interpretation of the topic, with ‘Habitat’, ‘Reactions’, ‘Language,’ ‘The Present’, and ‘Our Future’ all being underpinned by an emergent theme of the need to re-centre climate change around humankind.

In considering future communications around climate change, this study recommends that the role of humankind in causing and potentially mitigating climate change is made explicit, and that in doing so scientists and communication experts consider carefully the language that is being used. In particular, it is vital to determine if a monopoly of English and/or technical scientific language is at risk of de-personalising the topic, thereby making it less likely to be discussed.

This study has also outlined how poems might be used as a form of data to provide further insight into the interpretation of scientific topics by non-specialists, and how such interpretations might lead to recommendations to establishing a dialogue with such an audience. The main limitations of this method are via the potential for bias in either the selection of the poetry or in the coding and subsequent analysis. However, by selecting a broad range of independent poetry (as was done here) and taking care to outline the transparency of such an approach (for example by carefully describing the relationship between emergent codes, categories, and themes), the trustworthiness of this method can be established. In order to further explore the importance of language a future study that investigated the interpretation of poetry written in multiple languages and dialects would be conducive; however, such an interpretation would be reliant on a multilingual research team and/or translation of the poems that had been sanctioned by the poet.

At the beginning of the poem ‘Tip #5 What not to say whilst online dating‘ Helen Moore quotes the American poet political activist Grace Paley (Howard et al., 2018, p. 60):

> It is the responsibility of the poet to be a woman to keep an eye on this world and cry out like Cassandra, but be listened to this time.

In Greek mythology, Cassandra was the daughter of Priam and Hecuba and was cursed to utter prophecies that were true but that no one believed. Clearly this responsibility should not just lie with the poet, but in interpreting climate change for a non-specialist audience, the poets that featured in this study have demonstrated the importance of re-positioning humans at the very centre of the topic.

## Data Availability

The poems that were selected for the analysis, along with their coded categories, are available through (Illingworth, 2019; DOI: 10.17605/OSF.IO/T2YDR)

## Competing interests

Author SI is the chief executive editor of *Geoscience Communication*.

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

*Table 1: the codes that emerged from the content analysis. *The number of occurrences is not limited to one per poem.*

| Code | Description | Occurrences* |
|---|---|---|
| **Fauna** | Makes specific reference to mammals (other than humans), insects, birds, fish, etc. | 61 |
| **Flora** | Make specific reference to plants, trees, etc. | 32 |
| **Mutually Exclusive** | Makes specific reference to humans and nature being unable to live together in harmony. | 31 |
| **Science** | Makes specific reference to a specific scientific fact or piece of scientific information. | 31 |
| **Acknowledgment** | Makes specific reference to acknowledging that there is something wrong with the current climate system. | 30 |
| **Humans** | Makes specific reference to humans, not as the narrator of the poem but rather as objects that feature in it. | 28 |
| **Weather** | Makes specific reference to the weather. | 26 |
| **Blame** | Specifically attribute blame to someone / something for the current state of the climate system. | 22 |
| **Death** | Makes specific reference to death. | 19 |
| **Spiritual** | Makes specific reference to a spiritual or religious concept. | 19 |
| **Children** | Makes specific reference to children. | 16 |

| | | |
|---|---|---|
| **Other Language** | Used another language (other than English) to communicate what they wished to express. | 14 |
| **Pollution** | Makes specific reference to pollution. | 11 |
| **Hope** | Makes specific reference to hope that is either present in or may arise from the current state of the climate system. | 10 |
| **Future** | Makes specific reference to the future. | 9 |
| **Looking Away** | Makes specific reference to humans looking away or being agnostic in our attitudes towards the current climate system. | 9 |
| **Broken** | Makes specific reference to things being broken. | 7 |
| **Doubt** | Makes specific reference to doubting the existence and impacts of negative anthropogenic climate change. | 6 |
| **Solutions** | Makes specific reference to a potential solution to the negative effects of climate change. | 4 |
| **Specific Event** | Makes reference to a specific event brought about / affected by climate change. | 2 |

*Table 2: the categories that emerged, alongside their corresponding codes. *The number of occurrences is not limited to one per poem.*

| **Category** | **Corresponding Codes** | **Occurrences*** |
|---|---|---|
| **Habitat** | Fauna, Flora, Mutually Exclusive, Humans | 152 |
| **Reactions** | Acknowledgment, Blame, Hope, Looking Away, Doubt, Solutions | 81 |
| **Language** | Science, Spiritual, Other Language | 65 |
| **The Present** | Weather, Death, Pollution, Broken, Specific Event | 65 |
| **Our Future** | Children, Future | 25 |