# Peer review of "This bookmark gauges the depths of the human"

_Geoscience Communication, 2019_

## Referee Comment (RC1) · Anonymous Referee #1 · 24 Sep 2019

Thank you very much for the opportunity to review this manuscript. I found the work to be particularly well written, and the topic especially timely. I also appreciated the author taking the time to set-out and give detail to the methodology that informed and framed this research - this is often a over-looked element of qualitative research that I believe is important with regards to interpreting the dependability of the study and allowing reviewers like me to follow, audit, and critique the study. I also found it particularly valuable to have lines from different poems represented within the text and in relation to the different codes identified through the study.

I really only have minor feedback that I would ask the author to consider.

Introduction

[Figure]

Starting on Line 79. I know the focus of the paper is on the nonexpert communicating about climate change, but as you also note in your methodology and evaluation of the author's of the poems in your study, sometimes scientists are poets, poets are scientists. I wonder if you could highlight, even if just briefly in this paragraph of the introduction, the potential value, indeed examples of, scientists who do communicate about climate change through poetry? This has been highlighted in a related paper as one way that they [scientists], and others, can communicate and generate dialogue about complex topics (see Januchowski-Hartley et al. 2018 and the text about scientists who write poems in relation to their research and even their interpretations of climate reports). Perhaps this only warrants a brief mention in the introduction, and potentially then also revisited in your conclusion section, as noted below. I believe it is important that we not make an unnecessary dichotomy between scientists and poets, and as you found in your study, these people do exist, and it is possible that even others who were not explicit about their professional life in their author bio are also scientists (here in the broadest sense).

Conclusion

Starting on Line 779. Here I think you could potentially highlight how poetry can be used as a venue/method/or conduit for diverse people, including scientists, to establish a dialogue amongst each other. The paper referenced above by Januchowski-Hartley et al. 2018 also highlighted the value for scientists, and indeed those learning science, to include poetry in their practice and lives to engage with the topics they work on (or learn about) and to communicate about those topics in broader dialogues. I noted above that it would be a shame to segregate out scientists, not all scientists are climate scientists either, from consideration of non-specialists. I do appreciate that you retained those people in the study who did self-identify as scientists.

Perhaps my above point also links into your closing paragraph. You could link to /cite related works where scientists, particularly climate scientists, are also using poetry (and visual artwork) to interpret climate change; that can be interpreted as being for

non-specialist audiences and move toward broadening the dialogue. I leave it for you to consider; I thought it potentially strengthens or broadens your closing argument.

Finally, I do see value in multiple colleagues undertaking the content analysis; even if conducting separate content analyses and then comparing the messages that emerge. Perhaps this is an additional direction that could be pursued in future works that you or others lead. This would be valuable in also recognizing different people's interpretations of poems, because after all, 'Do nothing to a poem that it never was written to have done to it' (Robert Frost), and whatever our interpretations are of a poem, are potentially not those of what the author intended. This could also suggest some potential value in a follow-up study that couples content analysis with interviews [of poets] (though I recognize some poets might not be comfortable with that).

Thank you again for the opportunity to read and contribute ideas and feedback on this manuscript. I hope that some of the ideas and feedback that I shard are useful for this or broader research that you go on to conduct.

---

## Referee Comment (RC2) · Anonymous Referee #2 · 27 Sep 2019

General Comments The article 'This bookmark gauges the depth of the human: How poetry can help to personalize climate change' is a highly interesting publication, connecting modern literary criticism with the current public debate about climate change. The focus is on analyzing 72 poems by various authors to identify how climate change is framed and represented in contemporary poetry. The underlying notion is that poems can bridge the divide between academic evidence and the still remote but needed change in personal lifestyle and attitude. This is in accordance with results from modern communication and social sciences which have shown that despite climate change is widely accepted, the human impact on the climate is still questioned. The author therefore seeks to find new ways to provoke the public to connect academic knowledge about climate change with individual actions on a very personal level. Poetry –

as other forms of arts such as theatre, music or painting – might be able to become a catalyzer for this transformation.

Specific Comments The role of emotions in science communication is not explicitly addressed but seems to be critical. Here, further reference to current research on the role of emotionality in science communication can increase the rationale for this article. (e.g. Smith & Leiserowitz (2014) The Role of Emotion in Global Warming Policy Support and Opposition. Risk Analysis. Vol 34 (5). Doi:10.1111/risa.12140) [Line 86 to 90] Stuart Hall's concept of 'Encoding – Decoding' can help to shed light on the central problem discussed in this publication. While the 'academic language' is used by scientists to convince decision makers to take action against climate change, this language is not successfully decoded by the public. Poetry can offer a different "language" that might help to decode climate change from another perspective. Hall – while being somewhat outdated – might present a theoretical frame for this article from a social or even cognitive science perspective and to introduce a somewhat more critical perspective on the interpretation of poetry as well. Hall, S.: Encoding/decoding in Television Discourse, in: Centre for Contemporary Cultural Studies: Culture, Media, Language: Working Papers in Cultural Studies, 1972–79, Hutchinson, London, 1973.

[Line 157 to 175] To better understand the sample, an introduction into the readership of the Magma magazine would be helpful. Otherwise, one might wonder about potential social-cultural biases or a moral framing effect (maybe even some sort of confirmation bias) related to the overall magazine's concept and marketing strategy.

Methodology and operationalization is very well described in chapters 2.3 and 2.4. Nevertheless, the description of the analytical method lacks reference to e.g. the thematic analysis approach, which has been critically described for example by Braun and Clark (Braun & Clarke (2006). Using Thematic Analysis in Psychology. Qualitative Research in Psychology Vol. 3 (2)).

[Line 406 to 412] A very lively discussion among anthropologists is addressed here

– the conception of nature and the role of humans within (or outside) this concept. This could be addressed by referring to e.g. Habermas (2004). The Future of Human Nature. or Descola (2013). Beyond Nature and Culture.

[Line 727 to 741] I highly appreciate the critical element in this chapter, but I may have missed the link to the analysis of the climate related poetry. While I fully support the statements in this paragraph, I would like to recommend a more robust transition from the analysis results to the statement proclaimed. Since 3.6 represents the core message of this article, a sound argumentation is needed to strengthen the claim, that "the central role occupied by humankind" can be derived from the poetry analyzed.

[Lines 769 to 770] I am surprised, that there is no category dedicated to the actors/main characters of the poems. Especially, while you argue that all categories are "underpinned by an emergent theme of the need to re-center climate change around humankind." Maybe you can briefly explain, while you have not focused on the actors?

No technical corrections required

---

## Referee Comment (RC3) · Anonymous Referee #3 · 30 Sep 2019

I can envisage there being value in an overview and analysis of poetry with relation to the environment that uses categorisation and similar procedures, perhaps along the lines of the 'distant reading' methodology of the Stanford Literary Lab; or, on another track entirely, an analysis of how poetry has been or can be used in public engagement contexts, or perhaps in self-conscious collaboration with scientists and/or communicative agendas. However, the sample of work here was too small to support the first endeavour, and the second did not seem to be at issue, though the model of communication which underpinned the essay suggested this as the most appropriate context. Broadly speaking, the article requires much more nuanced framing and discussion. Even given the journal's remit of raising awareness of the importance and value of science communication from a scientist's point of view, and understanding that poetry

is being examined within that context, the discussion here cannot avoid involving concepts, ideas and methods that are well-estabished in non-science fields, which bear on the discussion of poetry in any disciplinary or cultural context, and which are currently absent or insufficiently considered.

I am afraid that I found the discussion of poetry to be reductive, ahistorical and simplistic. What evidence is there for poetry being 'something that can transcend cultural barriers' (cf. issues of translation, cultural capital, marketing and publishing economies, etc), and why should poetry, any more than any other medium, be able to 'contextualize and personalise a global problem'? Particularly when one imagines the tiny readership for Magma and other poetry in comparison to other mediums! How does the fact that much poetry since at least the high modernist period has been criticized for being – and in some cases deliberately has been – difficult, oblique and non-referential, relate to the presentation of it as establishing a 'common language'? A claim which seems to unconsciously draw on Wordsworth's 1802 Preface to Lyrical Ballads ('a selection of language really used by men', etc), but struggles to account for much of the actual writing, publishing and reception of poetry since that time. A single issue of Magma is not sufficient to prove the overarching argument claimed – which would need to be revised to at least take into account the particular nature of that publication and of poetry magazine publication more broadly (readership, aesthetic, and so on). There exist many other collections of environmental poetry which would deepen the context for this argument, and also greatly complicate it (e.g. The Ground Aslant, ed. Harriet Tarlo; The Thunder Mutters: 101 Poems for the Planet ed. Alice Oswald). More incidentally, but perhaps still tellingly, Shelley's treatise was written in 1821 and published in 1840 (unlike your edition) – and the original historical context in which the poem was written goes a long way towards explaining its thinking and intent, which has since undergone, it is an understatement to say, considerable discussion, revision and contestation.

While the coding of poems by categories might potentially yield some useful analysis, I do not think it is sophisticated or subtle enough here to answer 'RQ1: how have poets

interpreted the, at times, esoteric principles of climate change?' (140). Perhaps it is simply a case that the RQ needs rephrasing, but there are basic questions here that are being conflated, perhaps the most pressing of which is: can the poets' interpretations of climate change (and surely the more appropriate word would be something like 'renderings' or 'representations of') be assumed to be identical with those of readers? And as the answer is surely 'no', where does that leave the communication argument? Complex questions of poetic functioning, representition and of reading/interpretation are being overridden.

It is unclear to me whether sections of poems could be and were multiply categorized. For instance, 'But these fields are, / again, under water, brought / to the brink of drowning' was mentioned for being categorized as 'the present', but is it not also 'reaction' and 'habitat'? More broadly, the categorizing needs to be much tighter and more targeted to be operable. For instance, 'Reactions', 'those poems that explore the reactions that humans have towards climate change' – it is hard to see how any poem dealing, however tangentially, with climate change wouldn't fall into this category? The positioning of the extracts from the poems narrows down the possible complexity of the questions under discussion, and of the extracts themselves. A minor instance: the author states that poets 'had clearly chosen to write sections of the poem in a language other than English as it enabled them to more fully express what it was that they meant to say about climate change', but other possible reasons can surely be envisaged (e.g. questions of cultural capital, identity formation, deliberate estrangement of Anglophone reader etc.).

The conclusions reached were rather anticlimactic and commonplace. For instance, is it news to anyone that 'using only a singular official language (i.e. English) or technical language (i.e. science) is not sufficient to interpret and communicate the causes and consequences of climate change, and that by doing so we are at risk of ostracising those communities that are not fluent in these chosen languages' (564-8)? The question of communication is reduced to the overly narrow purview of issues such as

language (which is in any case too casually categorized and understood – there are very many theories of poetic language which needed to be taken into account here, e.g. Jakobson's Functions of Language, 1960, itself much contested since) and subject matter; and more consideration surely needs to be given to questions of ideology and its formation and perpetuation, within with communication takes place. The idea that climate change 'is discussed less widely than is needed for meaningful action to take place' (572-3) obscures the fact that climate change is surely discussed very widely and with great frequency (see any newspaper), and the implication that more meaningful action awaits better communication needs at least some reflection and justification, and probably qualification.

---

## Author Comment (AC1) · 31 Oct 2019

Thank you for taking the time to read this manuscript, and for providing helpful and specific feedback for how to improve this work. Below I have responded to all your comments (which for ease of use I have **written in bold**), and indicated how I have changed the manuscript to account for these changes. Any line references refer to those provided in the *Geoscience Communication Discussions* preprint.

**Thank you very much for the opportunity to review this manuscript. I found the work to be particularly well written, and the topic especially timely. I also appreciated the author taking the time to set-out and give detail to the methodology that informed and framed this research - this is often a over-looked element of**

[Figure]

**qualitative research that I believe is important with regards to interpreting the dependability of the study and allowing reviewers like me to follow, audit, and critique the study. I also found it particularly valuable to have lines from different poems represented within the text and in relation to the different codes identified through the study.**

Thank you for such a generous and kind comment. It is very heartening to hear that this research is appreciated, and it encourages me to continue to pursue this line of work in my future research.

**Starting on Line 79. I know the focus of the paper is on the nonexpert communicating about climate change, but as you also note in your methodology and evaluation of the authors of the poems in your study, sometimes scientists are poets, poets are scientists. I wonder if you could highlight, even if just briefly in this paragraph of the introduction, the potential value, indeed examples of, scientists who do communicate about climate change through poetry? This has been highlighted in a related paper as one way that they [scientists], and others, can communicate and generate dialogue about complex topics (see Januchowski-Hartley et al. 2018 and the text about scientists who write poems in relation to their research and even their interpretations of climate reports). Perhaps this only warrants a brief mention in the introduction, and potentially then also revisited in your conclusion section, as noted below. I believe it is important that we not make an unnecessary dichotomy between scientists and poets, and as you found in your study, these people do exist, and it is possible that even others who were not explicit about their professional life in their author bio are also scientists (here in the broadest sense).**

Thank you for raising this important issue. It is of course very important to highlight that several scientists also write poetry and that these two identities are not mutually exclusive. In order to better address this point, I have inserted the following lines of text in the manuscript after Line 90 (in the Introduction):

The purpose of this research is not to introduce a mutual exclusivity between scientists and poets, as there are many examples of scientists for whom poetry is an integral part of their practice (Illingworth, 2019), and who do a commendable job of communicating their research (and the research of others) through poetry (see e.g. McCarty, 2014;Januchowski-Hartley et al., 2018 and references therein). Rather, this research seeks to investigate how poetry (as opposed to science) has been used to interpret climate change, and how this might then be used to re-consider the ways in which science also engenders dialogue around this topic.

**Starting on Line 779. Here I think you could potentially highlight how poetry can be used as a venue/method/or conduit for diverse people, including scientists, to establish a dialogue amongst each other. The paper referenced above by Januchowski-Hartley et al. 2018 also highlighted the value for scientists, and indeed those learning science, to include poetry in their practice and lives to engage with the topics they work on (or learn about) and to communicate about those topics in broader dialogues. I noted above that it would be a shame to segregate out scientists, not all scientists are climate scientists either, from consideration of non-specialists. I do appreciate that you retained those people in the study who did self-identify as scientists.**

This is a very important point, as poetry is indeed a very powerful conduit for establishing dialogue between diverse people, including between scientists and non-scientists. This has been explored in several of my other research papers (see e.g. Illingworth and Jack, 2018;Illingworth et al., 2018), which I also reference in the Introduction to this manuscript. However, the purpose of this study was not to investigate the potential for poetry to act as an active conduit, but rather to investigate how poets (who were mainly non-scientists) have interpreted the, at times, esoteric principles of climate change. Therefore, whilst I absolutely agree with your statement (and indeed base much of my research ethos on this), I believe that in this instance including a further exploration of this would be extending beyond the research design of this particular study.

**Perhaps my above point also links into your closing paragraph. You could link to /cite related works where scientists, particularly climate scientists, are also using poetry (and visual artwork) to interpret climate change; that can be inter- preted as being for non-specialist audiences and move toward broadening the dialogue. I leave it for you to consider; I thought it potentially strengthens or broadens your closing argument.**

Again, I absolutely agree with this point and whilst it is not the main focus of this study it is certainly worth highlighting, as such the following text has been inserted into the manuscript at Line 790:

Such future studies might also consider poetry that is being written by scientists to help interpret climate change, for example the work of Rachel McCarthy (McCarthy, 2015). This approach would also be conducive in helping to dismiss the notion that poetry and science are mutually exclusive rather than complementary fields of research and practice.

**Finally, I do see value in multiple colleagues undertaking the content analysis; even if conducting separate content analyses and then comparing the messages that emerge. Perhaps this is an additional direction that could be pursued in future works that you or others lead. This would be valuable in also recognizing different people's interpretations of poems, because after all, 'Do nothing to a poem that it never was written to have done to it' (Robert Frost), and whatever our interpretations are of a poem, are potentially not those of what the author intended. This could also suggest some potential value in a follow-up study that couples content analysis with interviews [of poets] (though I recognize some poets might not be comfortable with that).**

I am in complete agreement that multiple colleagues undertaking the content analysis would be of benefit for future research direction. As such I have inserted the following text into the manuscript directly after Line 787:

Future studies would also benefit from multiple colleagues undertaking the content analysis that has been described in this paper, as doing so would better recognise potential differences in any interpretations, thereby improving the triangulation of the coding and subsequent analysis.

**References**

Illingworth, S., Bell, A., Capstick, S., Corner, A., Forster, P., Leigh, R., Loroño Leturiondo, M., Muller, C., Richardson, H., and Shuckburgh, E.: Representing the majority and not the minority: the importance of the individual in communicating climate change, *Geosci. Commun.*, 1, 9-24, 10.5194/gc-1-9-2018, 2018.

Illingworth, S., and Jack, K.: Rhyme and reason-using poetry to talk to underserved audiences about environmental change, *Climate Risk Management*, 19, 120-129, https://doi.org/10.1016/j.crm.2018.01.001, 2018.

Illingworth, S.: *A sonnet to science: scientists and their poetry*, Manchester University Press, Manchester, UK, 2019.

Januchowski-Hartley, S. R., Sopinka, N., Merkle, B. G., Lux, C., Zivian, A., Goff, P., and Oester, S.: Poetry as a Creative Practice to Enhance Engagement and Learning in Conservation Science, *BioScience*, 68, 905-911, 2018.

McCarthy, R.: *Element*, Smith/Doorstop, Sheffield, UK, 2015.

McCarty, V. M.: Poetry, Science and Truth: The Case of' Poet-Scientists' Miroslav Holub and David Morley, *Interdisciplinary Science Reviews*, 39, 33-46, 2014.

---

## Author Comment (AC2) · 31 Oct 2019

Thank you for taking the time to read this manuscript, and for providing helpful and specific feedback for how to improve this work. Below I have responded to all your comments (which for ease of use I have **written in bold**), and indicated how I have changed the manuscript to account for these changes. Any line references refer to those provided in the *Geoscience Communication Discussions* preprint.

**The role of emotions in science communication is not explicitly addressed but seems to be critical. Here, further reference to current research on the role of emotionality in science communication can increase the rationale for this article. (e.g. Smith Leiserowitz (2014) The Role of Emotion in Global Warming Policy**

**Support and Opposition. Risk Analysis. Vol 34 (5). Doi:10.1111/risa.12140) [Line 86 to 90] Stuart Hall's concept of 'Encoding – Decoding' can help to shed light on the central problem discussed in this publication. While the 'academic language' is used by scientists to convince decision makers to take action against climate change, this language is not successfully decoded by the public. Poetry can offer a different "language" that might help to decode climate change from another perspective. Hall – while being somewhat outdated – might present a theoretical frame for this article from a social or even cognitive science perspective and to introduce a somewhat more critical perspective on the interpretation of poetry as well. Hall, S.: Encoding/decoding in Television Discourse, in: Centre for Contemporary Cultural Studies: Culture, Media, Language: Working Papers in Cultural Studies, 1972–79, Hutchinson, London, 1973.**

I agree that further reference to current research on the role of emotionality in science communication would help to strengthen the justification for this research. I have stopped short of using the suggested works to determine the theoretical frame for this article, as I believe that I have already provided a detailed description of the research design for this study. Whilst such a re-framing is beyond the scope of this current work, it is certainly something that would merit further investigation in a future study. As such I have inserted the following text after Line 787:

In considering how poetry might offer a different perspective to science in interpreting climate change and its effects, future studies might also wish to consider the role of emotions (see e.g. Smith and Leiserowitz, 2014;Roeser, 2012), particularly with respect to establishing a common language.

**[Line 157 to 175] To better understand the sample, an introduction into the readership of the *Magma* magazine would be helpful. Otherwise, one might wonder about potential social-cultural biases or a moral framing effect (maybe even some sort of confirmation bias) related to the overall magazine's concept and marketing strategy.**

This is a very good point. I have inserted the following text after Line 787 to address the potential social-cultural biases that this may introduce:

While the poetry that was used for this analysis was selected because of its broad range, there is a potential limitation introduced by the relative exclusivity of submitting to poetry journals such as *Magma*. While *Magma* does not charge poets for submitting to their magazine (as was the case for 'The Climate Change Issue'), this is not the case for other journals. Furthermore, submitting work to poetry journals requires a certain level of cultural literacy that may risk excluding a range of diverse voices from contributing.

**Methodology and operationalization is very well described in chapters 2.3 and 2.4. Nevertheless, the description of the analytical method lacks reference to e.g. the thematic analysis approach, which has been critically described for example by Braun and Clark (Braun Clarke (2006). Using Thematic Analysis in Psychology. Qualitative Research in Psychology Vol. 3 (2)).**

Thank you for pointing this out, I agree that an additional reference could be provided here, and as such the following text has been added after Line 222:

A traditional approach to coding data during qualitative content analysis (see e.g. Braun and Clarke, 2006, and references therein) would be to begin by identifying meaning units in the text, condensing these down to smaller units and then labelling these units with codes.

**[Line 406 to 412] A very lively discussion among anthropologists is addressed here – the conception of nature and the role of humans within (or outside) this concept. This could be addressed by referring to e.g. Habermas (2004). The Future of Human Nature. or Descola (2013). Beyond Nature and Culture.**

Thank you for bringing my attention to these studies, and the references therein. I agree that my argument in this section would be strengthened by referring to this work,

and as such the following text has been inserted after Line 408:

This analysis supports the ongoing debate in anthropology about the conception of nature and the role of humans within this concept (see e.g. Descola, 2013;Habermas, 2014).

**[Line 727 to 741] I highly appreciate the critical element in this chapter, but I may have missed the link to the analysis of the climate related poetry. While I fully support the statements in this paragraph, I would like to recommend a more robust transition from the analysis results to the statement proclaimed. Since 3.6 represents the core message of this article, a sound argumentation is needed to strengthen the claim, that "the central role occupied by humankind" can be derived from the poetry analyzed.**

The emergence of "the central role occupied by humankind" came through a consideration of the five major categories that are discussed in Section 3.1 – 3.5 with respect to the RQs. The emergence of this theme is a result of the qualitative content analysis that I had described in Section 2.4, specifically Lines 226-228 and 284-286. In order to make this approach clearer I have inserted the following text after Line 286:

In determining these emergent themes, I re-considered each of the emergent categories with respect to the RQs, looking for any commonalities and/or overlaps, in a manner analogous to the emergence of the original codes and categories that is described above.

**[Lines 769 to 770] I am surprised, that there is no category dedicated to the actors/main characters of the poems. Especially, while you argue that all categories are "underpinned by an emergent theme of the need to re-center climate change around humankind." Maybe you can briefly explain, while you have not focused on the actors?**

I agree that exploring the actors of the poems would be interesting, and indeed in

my initial research design it is something that I had considered. However, I was not confident that I would be able to fully identify who the actors of the poems were in every instance, and that as such I would be introducing an additional degree of subjectivity that would potentially have weakened the reliability of the analysis. Future studies could certainly be aimed in this direction, perhaps aligned with either an interpretation of the poetry by multiple researchers (see 'Response to Referee 1') or a correspondence with the poets to more accurately represent the actors in the poems.

**References**

Braun, V., and Clarke, V.: Using thematic analysis in psychology, *Qualitative research in psychology*, 3, 77-101, 2006.

Descola, P.: *Beyond nature and culture*, University of Chicago Press, 2013.

Habermas, J.: *The future of human nature*, John Wiley Sons, 2014.

Roeser, S.: Risk communication, public engagement, and climate change: a role for emotions, *Risk Analysis: An International Journal*, 32, 1033-1040, 2012.

Smith, N., and Leiserowitz, A.: The role of emotion in global warming policy support and opposition, *Risk Analysis*, 34, 937-948, 2014.

---

## Author Comment (AC3) · 31 Oct 2019

Thank you for taking the time to read this manuscript, and for providing comments on how it could be improved. Below I have responded to all your comments (which for ease of use I have **written in bold**), and indicated how I have changed the manuscript to account for these changes. Any line references refer to those provided in the *Geoscience Communication Discussions* preprint.

**I can envisage there being value in an overview and analysis of poetry with relation to the environment that uses categorisation and similar procedures, perhaps along the lines of the 'distant reading' methodology of the Stanford Literary Lab; or, on another track entirely, an analysis of how poetry has been or can be used**

[Figure]

**in public engagement contexts, or perhaps in self-conscious collaboration with scientists and/or communicative agendas. However, the sample of work here was too small to support the first endeavour, and the second did not seem to be at issue, though the model of communication which underpinned the essay suggested this as the most appropriate context. Broadly speaking, the article requires much more nuanced framing and discussion. Even given the journal's remit of raising awareness of the importance and value of science communication from a scientist's point of view, and understanding that poetry is being examined within that context, the discussion here cannot avoid involving concepts, ideas and methods that are well-established in non-science fields, which bear on the discussion of poetry in any disciplinary or cultural context, and which are currently absent or insufficiently considered.**

I am sorry that you do not approve of the methodology that I adopted in this study. Naturally, as this is the first study of its kind I would expect there to be some criticisms of the approach that I have adopted. However, I believe that my methodology is carefully laid out and fully justified in the manuscript. I disagree that this article requires more nuanced framing and discussion, as what I have set out to do is to demonstrate how poetry might be analysed using qualitative content analysis, carefully outlying the limitations of my study, and suggesting how future endeavours might seek to build on and expand this work. Furthermore, as can be seen from the breadth of my references, this study has sought to fully engage with concepts, ideas and methods that are well-established in non-science fields.

**I am afraid that I found the discussion of poetry to be reductive, ahistorical and simplistic. What evidence is there for poetry being 'something that can transcend cultural barriers' (cf. issues of translation, cultural capital, marketing and publishing economies, etc), and why should poetry, any more than any other medium, be able to 'contextualize and personalise a global problem'? Particularly when one imagines the tiny readership for *Magma* and other poetry in com-**

**parison to other mediums! How does the fact that much poetry since at least the high modernist period has been criticized for being – and in some cases deliberately has been – difficult, oblique and non-referential, relate to the presentation of it as establishing a 'common language'? A claim which seems to unconsciously draw on Wordsworth's 1802 Preface to Lyrical Ballads ('a selection of language really used by men', etc), but struggles to account for much of the actual writing, publishing and reception of poetry since that time. A single issue of *Magma* is not sufficient to prove the overarching argument claimed – which would need to be revised to at least take into account the particular nature of that publication and of poetry magazine publication more broadly (readership, aesthetic, and so on). There exist many other collections of environmental poetry which would deepen the context for this argument, and also greatly complicate it (e.g. The Ground Aslant, ed. Harriet Tarlo; The Thunder Mutters: 101 Poems for the Planet ed. Alice Oswald). More incidentally, but perhaps still tellingly, Shelley's treatise was written in 1821 and published in 1840 (unlike your edition) – and the original historical context in which the poem was written goes a long way towards explaining its thinking and intent, which has since undergone, it is an understatement to say, considerable discussion, revision and contestation**

I apologise for any offence that I have caused in my discussion of the poetry in this research study, it was certainly not my intent to cause any ill harm.

With regards to the limitations of using a single issue of *Magma*, I believe that I have fully identified these limitations in the manuscript. However, as noted in my response to Referee 2, restricting this study to the poems that featured in 'The Climate Change Issue' does introduce a limitation to the study. I have now addressed this by inserting the following text after Line 787:

While the poetry that was used for this analysis was selected because of its broad range, there is a potential limitation introduced by the relative exclusivity of submitting to poetry journals such as *Magma*. While *Magma* does not charge poets for submitting

to their magazine (as was the case for 'The Climate Change Issue'), this is not the case for other journals. Furthermore, submitting work to poetry journals requires a certain level of cultural literacy that may risk excluding a range of diverse voices from contributing.

Thank you for your helpful comment regarding Shelley's *A Defence of Poetry*. The edition that I was using was from 1890 (not 1840), although I have amended the text so that the reader is fully aware of the providence of the text. The following text now appears in Line 72:

In his treatise A Defence of Poetry (written in 1821 and first published posthumously in 1840), the English Romantic poet P.B. Shelley (1890, pp. 46) wrote that:

**While the coding of poems by categories might potentially yield some useful analysis, I do not think it is sophisticated or subtle enough here to answer 'RQ1: how have poets interpreted the, at times, esoteric principles of climate change?' (140). Perhaps it is simply a case that the RQ needs rephrasing, but there are basic questions here that are being conflated, perhaps the most pressing of which is: can the poets' interpretations of climate change (and surely the more appropriate word would be something like 'renderings' or 'representations of') be assumed to be identical with those of readers? And as the answer is surely 'no', where does that leave the communication argument? Complex questions of poetic functioning, representation and of reading/interpretation are being overridden.**

Thank you for your comments, but what you are proposing is a completely different research project to the one that I have designed and carried out. I appreciate the time that you have taken in reading and critiquing this manuscript, but it is clear that I have not conducted a study in the way that you would have done yourself if you were also conducting a similar investigation. As such I must respectfully disagree with your comments, as we clearly have a fundamental difference of opinion with regards to the

research design that I have adopted, and which I have subsequently fully justified in the manuscript.

**It is unclear to me whether sections of poems could be and were multiply categorized. For instance, 'But these fields are, / again, under water, brought / to the brink of drowning' was mentioned for being categorized as 'the present', but is it not also 'reaction' and 'habitat'? More broadly, the categorizing needs to be much tighter and more targeted to be operable. For instance, 'Reactions', 'those poems that explore the reactions that humans have towards climate change' – it is hard to see how any poem dealing, however tangentially, with climate change wouldn't fall into this category? The positioning of the extracts from the poems narrows down the possible complexity of the questions under discussion, and of the extracts themselves. A minor instance: the author states that poets 'had clearly chosen to write sections of the poem in a language other than English as it enabled them to more fully express what it was that they meant to say about climate change', but other possible reasons can surely be envisaged (e.g. questions of cultural capital, identity formation, deliberate estrangement of Anglophone reader etc.).**

You have highlighted here what I agree is the main limitation of this study, i.e. that additional researchers conducting their own content analysis and creating their own codebooks would improve the triangulation of the analysis that I provide, and that multiple colleagues undertaking the content analysis would be of benefit for future research direction. As such I have inserted the following text into the manuscript directly after Line 787:

Future studies would also benefit from multiple colleagues undertaking the content analysis that has been described in this paper, as doing so would better recognise potential differences in any interpretations, thereby improving the triangulation of the coding and subsequent analysis.

**The conclusions reached were rather anticlimactic and commonplace. For instance, is it news to anyone that 'using only a singular official language (i.e. English) or technical language (i.e. science) is not sufficient to interpret and communicate the causes and consequences of climate change, and that by doing so we are at risk of ostracising those communities that are not fluent in these chosen languages' (564-8)? The question of communication is reduced to the overly narrow purview of issues such as language (which is in any case too casually categorized and understood – there are very many theories of poetic language which needed to be taken into account here, e.g. Jakobson's Functions of Language, 1960, itself much contested since) and subject matter; and more consideration surely needs to be given to questions of ideology and its formation and perpetuation, within with communication takes place. The idea that climate change 'is discussed less widely than is needed for meaningful action to take place' (572-3) obscures the fact that climate change is surely discussed very widely and with great frequency (see any newspaper), and the implication that more meaningful action awaits better communication needs at least some reflection and justification, and probably qualification.**

Again, I apologise that the conclusions that I reached in this manuscript, were in your opinion 'commonplace' and 'anticlimactic'. I must once again respectfully disagree with your commentary, as I believe that throughout this manuscript I have clearly evidenced both the research design and the subsequent analysis. Furthermore, I believe that the findings of this study will be of genuine use to people who are communicating climate science to diverse audiences, and that furthermore (as discussed at length in the manuscript), that this study provides a sturdy framework for people wishing to adopt a similar approach to analysing poetry using such an approach in the future – the commentary from the other referees would suggest that there is value in this, although I fully understand that this is not an opinion that you share.

**References**

Shelley, P. B.: *A defense of poetry*, Ginn, 1890.